# Heuristic-Guided Privacy against Membership Inference Attacks in Federated Learning

## Abstract

Federated Learning (FL) enables collaborative model training among several participants, while keeping local data private at the participants' premises. However, despite its merits, FL remains vulnerable to privacy attacks, and in particular, to membership inference attacks that allow adversaries to deduce confidential information about participants' training data. In this paper, we propose DINAR, a novel privacy-preserving FL method. DINAR follows a fine-grained approach that specifically tackles FL neural network layers that leak more private information than other layers, thus, efficiently protecting the FL model against membership inference attacks in a non-intrusive way. And in order to compensate for any potential loss in the accuracy of the protected model, DINAR combines the proposed fine-grained approach with adaptive gradient descent. The paper presents our extensive empirical evaluation of DINAR, conducted with six widely used datasets, four neural networks, and comparing against five state-of-the-art FL privacy protection mechanisms. The evaluation results show that DINAR reduces the membership inference attack success rate to reach its optimal value, without hurting model accuracy, and without inducing computational overhead. In contrast, existing FL defense mechanisms incur an overhead of up to +36% and +3,000% on respectively FL client-side and FL server-side computation times.

## 1 Introduction

Nowadays, advancements in Machine Learning (ML), along with the need of better privacy, have given rise to Federated Learning (FL) paradigm. FL enables collaborative model training among decentralized participants' devices, while keeping local data private at the participants' premises (McMahan et al., 2017). Thus, participants contribute by training their respective local models using their private data, and only transmit their local model parameters to a FL server, which aggregates these parameters to produce a global model. FL has various applications, such as e-health monitoring (Wu et al., 2022), disease diagnosis (Li et al., 2019), and fraud detection in banking systems (Grimm et al., 2021). Despite the privacy benefits offered by FL, recent studies have highlighted the vulnerability of FL systems to privacy inference attacks (Nasr et al., 2019; Lyu et al., 2020). These attacks, particularly Membership Inference Attacks (MIAs) (Shokri et al., 2017), exploit the parameters of the shared models to infer sensitive information about the training data of other participants. In a white-box FL configuration, where the model architecture and parameters are known to all participants, membership inference attacks pose a significant threat to privacy. For instance, an attacker on the server-side could discern from the aggregated parameters whether a specific individual's data was included in the training process. Similarly, a malicious participant on the client-side could deduce whether the data of a particular individual was used for training and potentially uncover sensitive information.

To address these privacy concerns, various FL defense mechanisms have been proposed (Abadi et al., 2016; Naseri et al., 2020; Papernot et al., 2018; 2017). These mechanisms leverage techniques such as cryptographic methods and secure multiparty computation (Zhang et al., 2019; Xu et al., 2019; Chen et al., 2021), trusted execution environments (Lebrun et al., 2022; Messaoud et al., 2022), perturbation-based methods and differential privacy (Naseri et al., 2020; Sun et al., 2019). Software and hardware-based cryptographic solutions provide interesting theoretical privacy guarantees, although at the expense of high computational overhead. Whereas existing perturbation-based methods negatively impact model utility and quality. Thus, our objective is to precisely strike

a balance between FL model privacy, model utility and computational cost for enabling effective privacy-preserving FL, especially in the case of cross-silo FL systems (*e.g.*, banking systems, hospitals, etc.) where the FL server shares the global model with the participating clients and not with external parties.

In this paper, we propose DINAR, a fine-grained privacy-preserving FL method that tackles membership inference attacks. This approach is motivated by an interesting observation made in recent studies (Mo et al., 2021a;b), and confirmed in our empirical analysis in §3, that is there is a layer in neural networks that leaks more private information than other layers. Thus, DINAR is based on a simple yet effective approach that consists in protecting more specifically the FL model layer that is the most sensitive to membership privacy leakage.

DINAR runs at the FL client-side, and allows to protect both the global FL model and the client models. Whereas for its own model predictions the client uses its privacy sensitive layer as part of the model, that privacy sensitive layer is obfuscated before sending client model updates to the FL server. Thus, the aggregated model produced by the FL server includes an obfuscated version of the privacy sensitive layer. And when the client receives the protected global model from the server, it first restores its local privacy sensitive layer (*i.e.*, the non-obfuscated version of that layer) that was stored during the previous FL round, and integrates it into its copy of the global model, before actually using the resulting personalized model for client predictions. Furthermore, in order to improve the utility of the protected model, DINAR leverages the adaptive gradient descent technique to further maximize the accuracy of the model (Duchi & Hazan, 2011). Indeed, given the high-dimensional nature of optimization problems in neural networks, adaptive gradient descent allows to dynamically adjust the model learning rate for each dimension in an iterative manner.

In particular, the paper makes the following contributions:

- We present an empirical analysis on several datasets and neural networks to characterize how much each layer of a neural network leaks membership privacy information.
- To the best of our knowledge, we propose the first fine-grained FL privacy-preserving method against MIAs, that specifically obfuscates the most privacy sensitive layer, for an effective yet non-intrusive privacy protection.
- We conduct extensive empirical experiments of our solution DINAR with six widely used datasets and four neural networks. We also compare DINAR against five state-of-the-art FL privacy protection mechanisms. Our evaluation results show that DINAR reduces the membership inference attack success rate to reach its optimal value, without hurting model accuracy, and without inducing computational overhead. In contrast, existing FL defense mechanisms incur an overhead of up to +36% and +3,000% on respectively client-side and server-side computation times.

## 2 BACKGROUND AND RELATED WORK

### 2.1 FEDERATED LEARNING

In Federated Learning (FL), instead of sharing raw data, several clients collaboratively train a predictive model. This allows better privacy, since clients only share their local model parameters to a server, that orchestrates the FL distributed protocol, while clients keep their raw data private on their devices. At each FL round, clients that are selected by the FL server to participate to that round train their local models using their own data. They then transmit their model updates to the FL server, which aggregates them to produce a new version of the global model shared with the clients. The classical algorithm used for model aggregation is FedAvg (Li et al., 2020b), a weighted averaging scheme that considers the amount of data each client has when aggregating the model updates. Furthermore, we consider the case where the FL server shares the global model with the participating clients and not with external parties. This is usually the case in cross-silo FL systems such as in banking systems, or between medical centers.

### 2.2 MEMBERSHIP INFERENCE ATTACK THREAT MODEL

Membership inference attacks (MIAs) aim to infer whether a data sample has been used to train a given model. Such attacks exploit vulnerabilities in the parameters and statistical properties of

the trained model to reveal information about the training data. Thus, it is important to safeguard individuals' confidentiality against MIAs that cause significant privacy violations, in particular, in areas involving highly sensitive information such as health applications, financial systems, etc.

**Attacker's Objective and Capabilities.** We consider the standard setting of a membership inference attack and its underlying attacker's capabilities Shokri et al. (2017). The objective of the attacker is to determine whether a given data sample was used by other participants for model training. An attacker can be on the client-side or the server-side. If the attacker is on the client-side, its goal is to determine, based on the global model, whether a data sample has been used for training by other clients, without knowing to which client it actually belongs. If the attacker is on the server-side, it is also able to determine, based on a client model, whether a data sample has been used by that client for training. The attacker has access to the parameters of the model for which it tries to violate privacy.

## 2.3 RELATED WORK

Cutting-edge research in countering membership inference attacks has made significant strides through innovative approaches, encompassing cryptographic techniques, secure hardware, and perturbation-based methods as summarized in Appendix B - Table 3. Cryptography-based solutions such as PEFL (Zhang et al., 2019), HybridAlpha (Xu et al., 2019), Chen et al. (Chen et al., 2021), or Secure Aggregation (Zheng et al., 2022), offer robust privacy solutions, with interesting theoretical guarantees. However, they tend to incur high computational costs due to complex encryption and decryption processes. Furthermore, these solutions often protect either the client-side or the server-side model, but not both, leaving potential vulnerabilities in the other unprotected component. Interestingly, solutions based on Trusted Execution Environments (TEEs) emerge as another alternative for better privacy protection (Lebrun et al., 2022; Messaoud et al., 2022). However, because of the high dimension of underlying models, striking a tight balance between privacy and computational overhead remains challenging.

On the other hand, perturbation methods, such as differential privacy (DP) with algorithm-specific random noise injection, serve as interesting safeguards against potential information leakage. When applied in the context of FL, DP has two main forms, namely Local Differential Privacy (LDP) that applies on client model parameters before transmission to the FL server (Chamikara et al., 2022), and Central Differential Privacy (CDP) where the server applies DP on aggregated model parameters before sending the resulting model to the clients (Naseri et al., 2020). WDP applies norm bounding and Gaussian noise with a low magnitude, which provides a good model utility (Sun et al., 2019). However, attack mitigation is limited, whereas computational costs. Recent works, such as PFA (Liu et al., 2021), MR-MTL (Liu et al., 2022), DP-FedSAM (Shi et al., 2023), and PrivateFL (Yang et al., 2023), allow better privacy and model utility. However, in practice, existing DP-based FL methods can only effectively improve privacy at the expense of utility, as shown in the evaluation presented later in the paper. Another approach to counter inference attacks in FL is through gradient compression techniques, since such techniques reduce the amount of information available for the attacker (Fu et al., 2022). However, such techniques also decrease the model utility. In summary, existing FL privacy-preserving methods tackling MIAs either rely on cryptographic techniques and secure environments which induce a high computational overhead, or reduce model utility and quality with classical perturbation-based methods. In contrast, we propose a novel method that follows a finer-grained approach applying obfuscation on specific parts of model parameters that leak privacy sensitive information. This results in good privacy protection, good model utility, and no perceptible computational overhead.

## 3 MOTIVATION OF A FINE-GRAINED PRIVACY-PRESERVING APPROACH

Recent studies analyzed the sensitivity and privacy risk of neural networks at a fine-grained level, to better characterize how much each layer of the model leaks privacy information (Mo et al., 2021a;b). As claimed in these studies, a similar pattern appears in all models, namely, there is a layer that leaks more private information than other layers. To better illustrate this behavior, we conduct an empirical analysis with four different datasets (GTSRB, CelebA, Texas100, Purchase100) and their

underlying models, deployed in a FL setting[1]. More precisely, we aim to characterize how much each layer of a model helps an attacker conducting MIAs to better infer if a given data sample was member of the model training set or not. In other words and as described in §2.2, such an attacker is able to differentiate between member data samples and non-member data samples. Thus, using a trained FL model, we conduct, on the one hand, a set of predictions with member data samples, and on the other hand, another set of predictions with non-member data samples. We then compute the gradients of each layer resulting from the predictions of member samples, and the gradients of each layer resulting from the predictions of non-member samples. Finally, we compute the generalization gap of each layer, *i.e.*, the difference between the gradients of member data samples and the gradients of non-member data samples. Thus, the higher the generalization gap, the more successful MIA is, *i.e.*, the easier it is for the MIA to differentiate between members and non-members, as shown in recent studies (Li et al., 2020a; Wu et al., 2023). Our empirical results are presented in Figure 1, where the generalization gap is computed using the widely used Jensen-Shannon divergence (Menéndez et al., 1997). We observe that different layers of a model may exhibit different generalization gaps. We also observe a similar behavior in all the model architectures, namely, the generalization gap of the penultimate layer is notably higher than the generalization gap of the other layers. Thus, that layer leaks more privacy sensitive information (*i.e.*, membership-related information), as shown in other studies (Mo et al., 2021a;b).

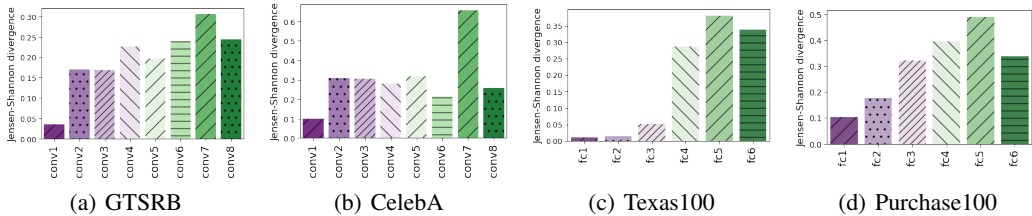

| (a) GTSRB | (b) CelebA | (c) Texas100 | (d) Purchase100 |

Figure 1: Layer-level analysis of divergence between member data samples and non-member data samples, using Jensen-Shannon divergence, when FL models are not protected against MIAs – FL models of GTSRB and CelebA include eight convolutional layers and one fully connected layer , and FL models of Texas100 and Purchase100 have six fully connected layers

## 4 DESIGN PRINCIPLES OF DINAR

We propose DINAR[2], a novel FL scheme for privacy protection against MIAs. The objective of DINAR is threefold: (i) improving resilience of models against MIAs, (ii) preserving model utility, and (iii) avoiding additional computational overheads. Whereas existing privacy-preserving FL methods either apply perturbation on all model layers, or use cryptographic techniques and secure environments, which induce a high computational overhead (as shown in §2.3 and §5.4), the intuition behind DINAR is to specifically handle the most *privacy sensitive layer* of a FL model, *i.e.*, the layer which reveals more client's privacy information than the others. This allows a non-intrusive yet effective solution to protect FL models against MIAs. DINAR is based on a prior knowledge of the most privacy sensitive layer of a model architecture, either based on prior studies such as (Mo et al., 2021a;b), or based on an empirical analysis of model architectures and datasets such as in Figure 1.

DINAR runs at the client-side, for each FL client that wants better protection against MIAs. Each DINAR instance on a client runs independently from the other clients' DINAR instances, and the interaction between the FL server and the clients follows the classical FL protocol, where at each FL round the clients send their local model updates to the server, and the server sends the aggregated global model to

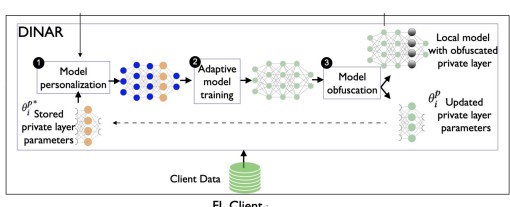

Figure 2: DINAR pipeline

---

[1]A description of the datasets, models and experimental setup can be found in §5.1.

[2]DINAR: fine-graine**D** pr**I**vacy-preservi**N**g feder**A**ted lea**R**ning

the clients. DINAR pipeline is presented in Figure 2, and consists of the successive stages of client model personalization, adaptive model training for improving model utility, and model obfuscation, that are detailed in the following. Furthermore, Algorithm 1 describe the different steps of DINAR pipeline.

---

**Algorithm 1:** DINAR algorithm on FL Client$_i$

**Inputs:** $\theta$: global model parameters; $p$: private layer index
**Output:** $\theta_i$: client model parameters

**Local variables:**
$\theta_i^{p}*$: parameters of private layer of client model
$(B^i, Y) = \{(B_1^i, Y_1), \ldots, (B_x^i, Y_x)\}$: training batches of Client$_i$
$\eta$ : learning rate

1 **Model Personalization**
2 **for** $j$ in $\{1..J\}$ **do**
3     **if** $j \neq p$ **then**
4        $\theta_i^j \leftarrow \theta^j$ // Use $j^{th}$ layer parameters from global model
5     **else**
6        $\theta_i^j \leftarrow \theta_i^{p}*$ // Restore parameters of client's private layer

7 **Adaptive Model Training**
8 $G \leftarrow 0$ // Set initial accumulated gradients matrix
9 **foreach** local training epoch **do**
10     **foreach** $(B_k^i, Y_k) \in (B^i, Y))$ **do**
11        $\widehat{Y}_k \leftarrow \theta_i(B_k^i)$ // Perform local prediction
12        $loss \leftarrow \mathcal{L}(Y_k, \widehat{Y}_k)$ // Compute model loss
13        $G \leftarrow G + \nabla_\theta.loss^2$ // Compute new cumulated gradients
14        $\theta_i \leftarrow \theta_i - \eta \frac{\nabla_\theta.loss}{\sqrt{G}+1e-5}$ // Update local model

15 **Model Obfuscation**
16 $\theta_i^{p}* \leftarrow \theta_i^p$ // Save parameters of client's private layer
17 $\theta_i^p \leftarrow$ random_values // Obfuscate parameters of client's private layer
18 **return** $\theta_i$

---

## 4.1 MODEL OBFUSCATION

In the following, we consider a model $M$ with $J$ layers, and model parameters $\theta$, where $\theta^1 \ldots \theta^J$ are the parameters of the respective layers $1 \ldots J$. We denote $p$ the index of the privacy sensitive layer of model $M$. At each FL round, Client$_i$ that participates to that round updates its model parameters $\theta_i$ through local training. Before sending the local model updates to the FL server, the client obfuscates the privacy sensitive layer of its model, namely $\theta_i^p$ that is the client model parameters of layer $p$. This obfuscation can be performed by applying differential privacy on $\theta_i^p$ layer (*i.e.*, DINAR/DP), or by simply replacing the actual value of $\theta_i^p$ by random values (*i.e.*, DINAR). The resulting local model updates are sent to the FL server for aggregation. Note that the raw parameters of the privacy sensitive layer (*i.e.*, before obfuscation) are stored at the client side in $\theta_i^{p}*$, and will be used in other stages of the DINAR pipeline.

## 4.2 MODEL PERSONALIZATION

As a first step of DINAR pipeline, when Client$_i$ participates to a FL round, it first receives the parameters $\theta$ of the global model $M$. Here, $\theta^p$, *i.e.*, the model parameters of the privacy sensitive layer $p$, correspond to obfuscated values. Client$_i$ integrates to its local model parameters $\theta_i$ all global model layer parameters but the parameters $\theta^p$ of layer $p$. Instead, Client$_i$ restores for that layer $\theta_i^{p}*$, its previously stored and non-obfuscated local model parameters of layer $p$. Thus, while the global FL model is protected against MIAs, Client$_i$ makes use of an effective personalized local model. This allows client model's privacy sensitive information to remain protected, while client data still contributes to the overall improvement of the global model through collaborative training.

## 4.3 ADAPTIVE MODEL TRAINING

While DINAR's model obfuscation and model personalization tackle model privacy against MIAs, DINAR pipeline also includes a stage to improve model utility through adaptive learning. This relies on the optimization of the loss function, denoted as $\mathcal{L}$, for each Client$_i$ and its local model $M_i$. $\mathcal{L}$ represents the cumulative errors of the client model $M_i$ across its training and testing data batches. In order to minimize the loss function $\mathcal{L}$, client model parameters $\theta_i$ are updated at each local training epoch, given a learning rate $\eta$ (with $\eta \in [0, 1]$). The latter serves as a coefficient that scales the computed gradient values at each learning epoch. To address model convergence challenges, DINAR leverages the adaptive gradient descent technique, which mitigates the issues associated with local minima and saddle points (Duchi & Hazan, 2011). Firstly, when training intricate models like Convolutional Neural Networks (CNNs) over multiple iterations, adaptive gradient descent allows a deliberate convergence, exhibiting a slower learning rate compared to algorithms such as Adam and RMSProp, particularly during the initial iterations (Kingma & Ba, 2015b; Mukkamala & Hein, 2017). Secondly, given the high-dimensional nature of optimization problems in neural networks, this technique dynamically adjusts the learning rate for each dimension in an iterative manner, which

holds true for both Adam and Adagrad. However, Adagrad does not implement momentum Karim-ireddy et al. (2020) contrary to Adam; as suggested and empirically observed by Jin et al. (2022), integrating momentum to the optimization algorithm may cause client-drift because of low participation rates of client devices, and incure potential convergence issues. Such a behaviour may even be exacerbated while increasing the number of client participants and introducing non-IID distributions between participants.

## 4.4 ANALYTICAL INSIGHTS

Figure 3 puts into perspective two aspects of a fine-grained analysis of model layers for privacy purposes. On the one hand, Figure 3(a) shows how much one can determine the divergence between member data samples that were used for model training and non-member data samples, by analyzing the one or the other of model layers. In this case, we consider CelebA dataset with a model containing eight convolutional layers. On the other hand, Figure 3(b) presents the result of a fine-grained protection that obfuscates the one or the other of local model layers. We observe that obfuscating the layer that leaks more membership information is actually sufficient to reach the optimal protection of the overall client model against MIAs[3]. Whereas obfuscating other layers that leak less membership information is not sufficient for the protection of the overall client model. This is the basis of the heuristics provided by DINAR. Note that similar behavior is observed with other datasets and models, although not presented here due to space limitation.

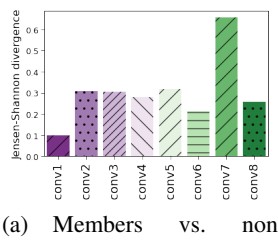

(a) Members vs. non-members

Furthermore, in order to provide an insight on DINAR's ability to preserve both privacy and model utility, we analyze the impact of DINAR and the considered baselines on the behavior of protected models. In Figure 4, we measure the loss of the attacked model separately for member and non-member data samples of Cifar-10, considering different defense methods. We evaluate the effectiveness of each defense technique in reducing loss distribution discrepancies between members and non-members, and in minimizing loss values. Ideally, the loss distribution of members and non-members should match, thus, resulting in model's lack of insightful information to distinguish members and non-members. First, we observe that in the no defense case, the loss distributions between members and non-members are very different, thus, enabling successful MIAs. DP-based techniques (i.e., LDP, CDP, WDP) reduce loss distribution discrepancies, however, at the expense of more frequent high loss values (i.e., lower model utility) due to noise added to all model layers' parameters. In contrast, a fine-grained obfuscation approach as followed by DINAR results in similar and more frequently low loss distributions of members and non-members, making MIAs more difficult and maintaining a good model utility.

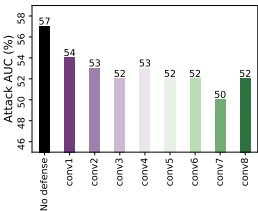

(b) Local model privacy

Figure 3: Per-layer analysis of divergence between members and non-member vs. resilience to MIAs with a fine-grained protection considering different obfuscated layers

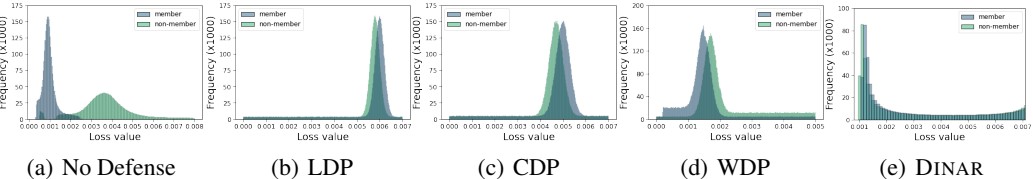

(a) No Defense        (b) LDP        (c) CDP        (d) WDP        (e) DINAR

Figure 4: Model loss distribution with different FL privacy-preserving techniques. The dark curve shows the loss distribution for member data samples, and the light curve shows the distribution for non-members

---

[3]50% is the optimal attack AUC that could be reached by a random protection approach, since determining the occurrence of a MIA is a binary decision.

## 5 EXPERIMENTAL EVALUATION

### 5.1 DATASETS AND EXPERIMENT SETUP

**Datasets.** We conduct experiments using a diverse set of datasets and models, encompassing four image datasets (Cifar-10, Cifar-100, GTSRB, and CelebA), a tabular dataset (Purchase100), and a raw audio dataset (Speech Commands). For each dataset, half of the data is used as the attacker's prior knowledge to conduct MIAs (Shokri et al., 2017), and the other half is partitioned into training (80%) and test (20%) sets. These datasets are sum up in Table 1, and further detailed in Appendix C.

**Baselines.** Our evaluation compares DINAR with different defense scenarios, including a no-defense baseline and five state-of-the-art solutions. Three of them, LDP (Chamikara et al., 2022), CDP (Naseri et al., 2020), and WDP (Sun et al., 2019), are inspired by Differential Privacy and employ various approaches for privacy preservation. We also include a cryptographic solution based on Secure Aggregation (SA) (Zheng et al., 2022) and another defense solution based on Gradient Compression (GC) (Fu et al., 2022).

These solutions, LDP, CDP, and WDP, employ various approaches for privacy preservation. For LDP and CDP, we set the privacy budget parameter $\epsilon = 2.2$ and the probability of privacy leakage $\delta = 10^{-5}$, following the findings of (Naseri et al., 2020). In the case of WDP, a norm bound of 5 is considered, and Gaussian noise with

Table 1: Used datasets and models

| Dataset | #Records | #Features | #Classes | Data type | Model |
|---|---|---|---|---|---|
| Cifar-10 | 50,000 | 3,072 | 10 | Images | ResNet20 |
| Cifar-100 | 50,000 | 3,072 | 100 | Images | ResNet20 |
| GTSRB | 51,389 | 6,912 | 43 | Images | VGG11 |
| CelebA | 202,599 | 4,096 | 32 | Images | VGG11 |
| Speech Commands | 64,727 | 16,000 | 36 | Audio | M18 |
| Purchase100 | 97,324 | 600 | 100 | Tabular | 6-layer FCNN |

a standard deviation of $\sigma = 0.025$ is applied. These settings ensure an optimal level of privacy preservation in our experiments. We also evaluate DINAR against a variant, DINAR/DP, that applies differential privacy to the private layer instead of generating random values for obfuscating that layer. As in DINAR, DINAR/DP personalizes FL client models (*c.f.*, §4.2). DINAR/DP uses the same hyper-parameters as LDP. Furthermore, for a fair comparison of model accuracy of the different defense methods, we also consider a variant (referred to with +) of each state-of-the-art defense method where adaptive model training is applied (*c.f.*, §4.3).

**Experimental setup.** The software prototype of DINAR is available: `https://anonymous.4open.science/r/dinar_87CD`.[4] All the experiments are conducted on an NVIDIA A40 GPU. We use PyTorch 1.13 to implement DINAR, and the underlying classification models. For the state-of-the-art defense mechanisms based on differential privacy, we employ the Opacus library (Yousefpour et al., 2021). We consider a FL system with 5 FL clients. Data are carefully divided into disjoint splits for each FL client, following a non-IID distribution. We run 50 FL rounds for Cifar-10, Cifar-100, GTSRB and CelebA, 80 FL rounds for Speech Commands, and 300 for Purchase100. We consider 5 local epochs per round for each FL client. Each dataset is splited into 80% for training, and 20% for testing. The learning rate is set to $10^{-3}$, and the batch size is 64. We evaluate FL privacy-preserving methods by measuring the attack AUC, as well as the model accuracy under maximum attack, and several cost-related metrics, as detailed in Appendix D.

### 5.2 EVALUATION OF PRIVACY PROTECTION

We first measure the effectiveness of DINAR in countering MIAs, *i.e.*, minimizing the attack AUC against both global and local models, considering Cifar-10 dataset. The attacker runs a white box membership inference attack, as described (Shokri et al., 2017). For each dataset and underlying model, we compare DINAR with defense baselines. We systematically evaluate DINAR, *WDP*, *LDP*, and *CDP* on both local and global models, considering the utility and the membership inference attack AUC. In Figure 5, we first plot distinctly the average attack AUC against local models. In all plots, each bar represents one defense scenario amongst the baselines we consider. Our results show that DINAR exhibits privacy mitigation rates that closely approach the 50% mark across all datasets, indicating a strong level of privacy protection. This holds true for both global and local model inference attacks, while differencial privacy mech-

---

[4]Anonymous link will be replaced by public git link in the camera-ready.

anisms are less constant at protecting the models. It is worth noting that DINAR achieves reducing the privacy leakage of local models by 19% in the best case, as shown by Figure 5, while differential privacy reveals its limits in that case: WDP only reduces the privacy leakage by 6% and in the best case, and even CDP is worse than DINAR by only reducing it by 15%.

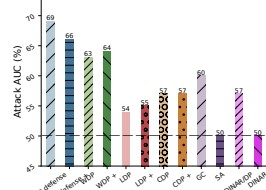

By concealing sensitive layers and replacing parameters by random values, DINAR enables the perturbation of the attacked model outputs as received by the attacker, thereby mitigating membership inference attacks. Indeed, the attacker receives an altered version of the model with randomized private layer parameters. And when the attacker tries to reproduce the behavior of the target model, the randomization necessarily impacts the outputs of the model, which makes them barely comparable to the outputs of the shadow model. This counters the logic of MIAs, and explains the drop of the attack AUC. When it comes to DINAR/DP, which provides protection by applying DP to the protected layer, we observe a lower protection than DINAR, due to the fact that the noisy data still shares some similarities with raw data. More extensive results with other datasets and models are presented in Appendix E, and corroborate our observations.

Figure 5: Privacy evaluation – The horizontal dashed line represents the optimal value of attack AUC (50%)

## 5.3 ANALYZING PRIVACY VS. UTILITY TRADE-OFF

With the objective of empirically confirming DINAR's ability to balance both privacy and model utility in a FL system, we evaluate its impact on local models behavior. We conduct the experiments on different datasets, by running the same attack scenario as the one presented in §5.2, introducing the consideration of both privacy and model utility metrics. Figure 6 shows our results by plotting both metrics on two axes: the x axis represents the average local model accuracy, while the y-axis plots the overall attack AUC we previously defined. In a best-case scenario, the dot should be located in the bottom-right corner of each plot, meaning that the effective defense mechanism both preserves the model accuracy and decreases the attack AUC to 50%. We observe that *WDP*, *CDP* and *LDP* achieve reasonable attack mitigation but often reduce model utility. For example, on the Purchase100 dataset, *WDP* reduces attack AUC by 2%, while *CDP* reduces it by 28%; however, with a significant reduction of model accuracy by 20%. DINAR/DP, which preserve a good model utility, is able to reduce the attack AUC by 10%, although, not fully protecting the model agains MIAs. In contrast, DINAR reaches the optimal attack AUC, with a model accuracy drop lower than 1%. In most cases, DINAR strikes a balance between privacy preservation and utility, demonstrating the effectiveness of mitigating membership information leakage in a fine-grained FL approach.

## 5.4 COST OF PRIVACY-PRESERVING MECHANISMS

In the following, we evaluate the possible overheads induced by privacy-preserving FL mechanisms across key metrics, such as model training duration at the client-side andmodel aggregation duration at the server-side. We additionally evaluate peak GPU memory usage for training and privacy protection at the client-side in Appendix K. In the following, we report on the costs of different defense mechanisms with the GTSRB dataset and VGG11 model, although other evaluations of overheads were conducted with other datasets and models, and resulted to similar observations.

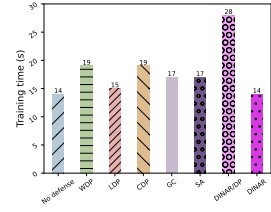

(a) Client-side

**Model Training Time.** We examine different scenarios to evaluate the average training duration per FL round for each client.

This duration refers to the total time required for all the local training epochs of a client during a FL round. The impact of privacy mechanisms like LDP, CDP, WDP, SC and SA on the training duration is depicted in Figure 7(a). We notice that incorporating privacy-preserving techniques that are based on differential privacy may have a negative effect on the overall training duration. Despite the improvements made by the Opacus framework in speeding up differential privacy, there is still a significant

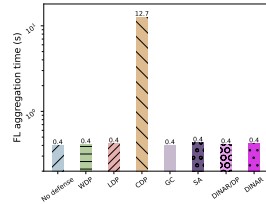

(b) Server-side

Figure 7: Computational cost of FL defense mechanisms

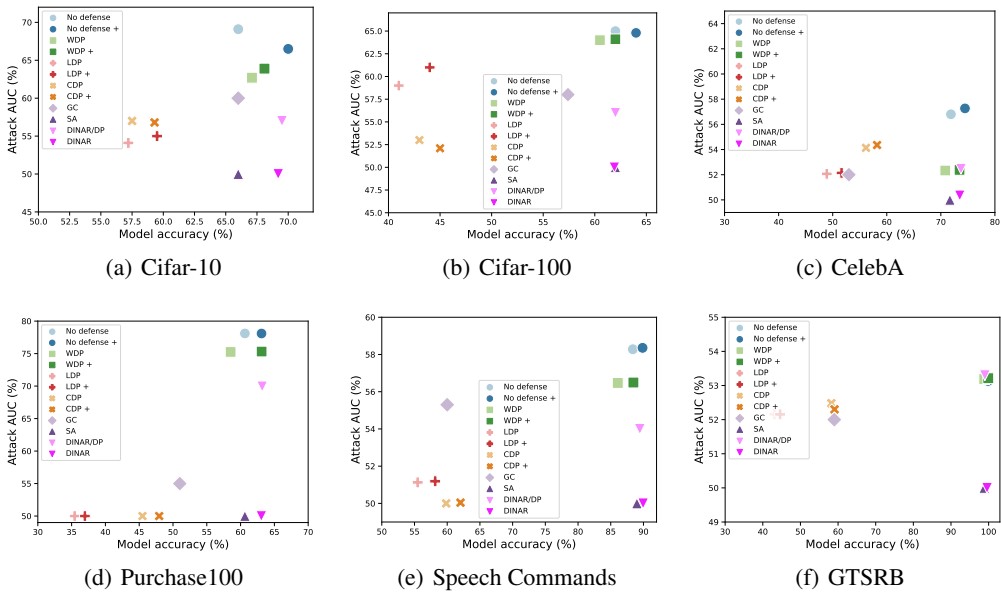

(a) Cifar-10      (b) Cifar-100      (c) CelebA

(d) Purchase100      (e) Speech Commands      (f) GTSRB

Figure 6: Trade-off between privacy and utility in different FL defense scenarios

cost. In the worst-case scenario, adding noise results in a training dura-
tion increased by 36%. However, it is important to highlight that DINAR
effectively addresses the computational overhead associated with differ-
ential privacy without compromising system performance.

**FL Aggregation Time.** We conduct measurements to determine the
average duration for server aggregation in various scenarios, that we plot in Figure 7(b). This in-
volved tracking the time taken from when the server received all weights to be aggregated until it
sent the aggregated weights. Notably, the use of CDP resulted in a significant increase in aggrega-
tion duration, reaching up to 30 times longer for GTSRB with VGG. This prolonged duration can be
attributed to CDP's design principle, which involves introducing noise to the parameter aggregate
before transmission to clients. This process substantially extends the time required for aggregation,
measured in seconds in our case. However, when employing DINAR, *LDP*, and *WDP*, the durations
exhibit similar orders of magnitude compared to the scenario without any baseline. This suggests
that these privacy mechanisms do not impose a substantial additional cost in terms of aggregation
time, presenting a more efficient alternative.

## 6 CONCLUSION

DINAR heuristics has the potential to better protect the privacy of FL systems against membership
inference attacks, both for global FL model and client models. DINAR follows a simple yet effec-
tive fine-grained approach that consists in protecting more specifically the model layer that is the
most sensitive to membership privacy leakage. This provides effective and non-intrusive FL privacy
protection. Furthermore, DINAR compensates for any potential loss in the accuracy of the protected
model by leveraging adaptive gradient descent and, thus, further maximizing model utility. Our
empirical evaluation using various widely used datasets, neural networks, and state-of-the-art FL
privacy protection mechanisms demonstrated the effectiveness of DINAR in terms of privacy, util-
ity and cost. Beyond FL defense against membership inference attacks, we envision that similar
fine-grained protection approaches could be used for other types of privacy attacks, such as property
inference attacks, and model inversion attacks. In addition, another interesting research direction to
explore is helping in automatically determining the neural network layers that are the most sensi-
tive to privacy leakage depending on the actual threat model, type of privacy attack, and FL model
architecture.

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

## A    APPENDIX – NOTATIONS

Table 2: Notations

| Notation | Description |
|---|---|
| N | Number of FL clients |
| $D_i$ | Local training data of $Client_i$ |
| $M$ | Global FL model |
| J | Number of layers in global FL model |
| $M_i$ | Local model of $Client_i$ |
| $\theta$ | Global model parameters |
| $\theta_i$ | Model parameters of $Client_i$ |
| $\theta^j$ | Parameters of the $j^{th}$ layer of global model |
| $\theta_i^j$ | Parameters of the $j^{th}$ layer of $Client_i$'s model |
| $p$ | The index of the private layer of a model, $e.g.$, $\theta_i^p$ are the parameters of the private layer of $M_i$ |
| $\theta_i^{p*}$ | Non-obfuscated parameters of private layer of $M_i$ stored on $Client_i$ |
| $\eta$ | Model learning rate |
| $\mathcal{L}$ | Model loss function |

## B    APPENDIX – SUMMARY OF RELATED WORK ON PROTECTION AGAINST MEMBERSHIP INFERENCE ATTACKS IN FEDERATED LEARNING

Table 3: Comparison of FL privacy-preserving methods against MIAs

| Privacy-preserving category | Protection method | Model protection against MIAs | Model utility | Negligible overhead |
|---|---|---|---|---|
| Cryptography-based methods | PEFL (Zhang et al., 2019) | ✓ | ✓ | ✗✗ |
| | HybridAlpha (Xu et al., 2019) | ✓ | ✓ | ✗✗ |
| | Chen et al. (Chen et al., 2021) | ✓ | ✓ | ✗✗ |
| | Secure Aggregation (Zheng et al., 2022) | ✓ | ✓ | ✗ |
| TEE-based methods | MixNN (Lebrun et al., 2022) | ✓ | ✓ | ✗✗ |
| | GradSec (Messaoud et al., 2022) | ✓ | ✓ | ✗✗ |
| Perturbation-based methods | CDP (Naseri et al., 2020) | ✓ | ✗ | ✗ |
| | LDP (Naseri et al., 2020) | ✓ | ✗ | ✗ |
| | FedGP Triastcyn & Faltings (2020) | ✓ | ✗ | ✗ |
| | WDP (Sun et al., 2019) | ✗ | ✓ | ✗ |
| | PFA (Liu et al., 2021) | ✗ | ✓ | ✗ |
| | MR-MTL (Liu et al., 2022) | ✗ | ✓ | ✗ |
| | DP-FedSAM (Shi et al., 2023) | ✗ | ✓ | ✗ |
| | PrivateFL (Yang et al., 2023) | ✗ | ✓ | ✗ |
| Gradient Compression | Fu et al. (Fu et al., 2022) | ✓ | ✓ | ✗ |
| *Our method* | DINAR | ✓ | ✓ | ✓ |

## C    APPENDIX – DETAILED DESCRIPTION OF DATASETS AND MODELS

**CelebA.** CelebFaces Attributes Dataset is a large face images dataset, with 202,599 images for facial recognition and attribute detection. A subset of 40,000 images, resized to 64x64 pixels, was randomly selected. We create 32 classes by combining five pre-annotated binary facial attributes (Male, Pale Skin, Eyeglasses, Chubby, Mouth slightly Opened) for each picture (Liu et al., 2015). The VGG11 architecture was employed for image processing (Simonyan & Zisserman, 2015).

**Cifar-10 and Cifar-100.** These are image dataset that consists of 60,000 images categorized into 10 classes for Cifar-10, and contains 100 classes for Cifar-100 (Krizhevsky et al., 2010). These datasets encompass a wide range of objects such as airplanes, automobiles, birds, cats, and more. Each image in these datasets has a resolution of 32x32 pixels. For our experiments, we employ the ResNet-20 model.

**Speech Commands.** This dataset is a Google-released audio waveform for speech recognition classification (Warden, 2018). It consists of 64,727 utterances from 1,881 speakers pronouncing 35 words (respectively 35 classes). Each audio record was transformed into a frequency spectrum with

a duration of 1 second. For classification, we use the M18 classifier, a convolutional model with 18 layers and 3.7M parameters (Dai et al., 2017).

**GTSRB.** German Traffic Sign Recognition Benchmark dataset comprises 51,389 records across 43 classes, specifically designed for traffic sign recognition. It captures real-world traffic scenarios, including variations in lighting, weather conditions, and camera angles. This dataset is widely used for evaluating traffic sign recognition algorithms and developing machine learning models for autonomous driving. We use VGG11 model architecture for this dataset (Houben et al., 2013; Simonyan & Zisserman, 2015).

**Purchase100.** It is a tabular dataset adapted from Kaggle's "Acquire Valued Shoppers" challenge, consisting of 97,324 records with 600 binary features representing customer purchases. The goal was to classify customers into 100 types based on their buying behavior Shokri et al. (2017). For modeling, we use a fully-connected neural network architecture with layers of sizes 4096, 2048, 1024, 512, 256, and 128, leveraging Tanh activation functions and a fully-connected classification layer (Jia et al., 2019).

## D  APPENDIX – DETAILED DESCRIPTION OF EVALUATION METRICS

**Attack AUC.** The attack success rate on a given model measures the percentage of successful MIAs conducted by an adversary. The attack AUC (Area Under the Curve) is a single value that measures the overall performance of the binary classifier implementing MIAs. The AUC value is within the range [50%–100%], where the minimum value represents the performance of a random MIA attacker, and the maximum value would correspond to a perfect attacker. The attack AUC is a robust overall measure to evaluate the performance of MIAs because its calculation involves all possible attacker's binary classification thresholds. Since the weakest (*i.e.*, most naive) MIA attacker would reach a minimum attack AUC of 50%, the best defense against MIAs would approach that optimal value of attack AUC of 50%. Thus, we use attack AUC as a means to evaluate the privacy of a model.

**Overall Model Privacy Metric.** In a FL system that consists of the global FL model $M$, and $N$ clients models $M_1 \ldots M_N$, we define a metric for measuring the overall privacy of all these models. Namely, we measure the highest potential privacy leakage from both the global model and clients' local models. Given the $F_{AUC}$ function for computing the attack AUC of a model, the overall model privacy of the FL system is computed as follows:

$$Max\left(F_{AUC}(M), \frac{\sum_{i=1}^{N} F_{AUC}(M_i)}{N}\right)$$

**Overall Model Utility Metric.** We evaluate the utility of a protected model by measuring its accuracy, namely the ratio of correctly classified instances to the total number of instances. Considering DINAR's approach for protecting FL clients' models, we consider the average of accuracy of clients' protected models. Given $N$ clients, $M_i$ the model of each $Client_i$, and $F_{Acc}$ the function that calculates accuracy of a model, the overall model utility metric is as follows:

$$\frac{\sum_{i=1}^{N} F_{Acc}(M_i)}{N}$$

**Cost-Related Metrics.** We also evaluate the additional costs that can be induced by a privacy-preserving FL mechanism, both in terms of execution times and memory usage. For instance, we measure the necessary time for a client to train a model during a FL round. We also measure the necessary time for the FL server to perform aggregation of client model updates. Finally, we measure the memory used by a client during model training,.

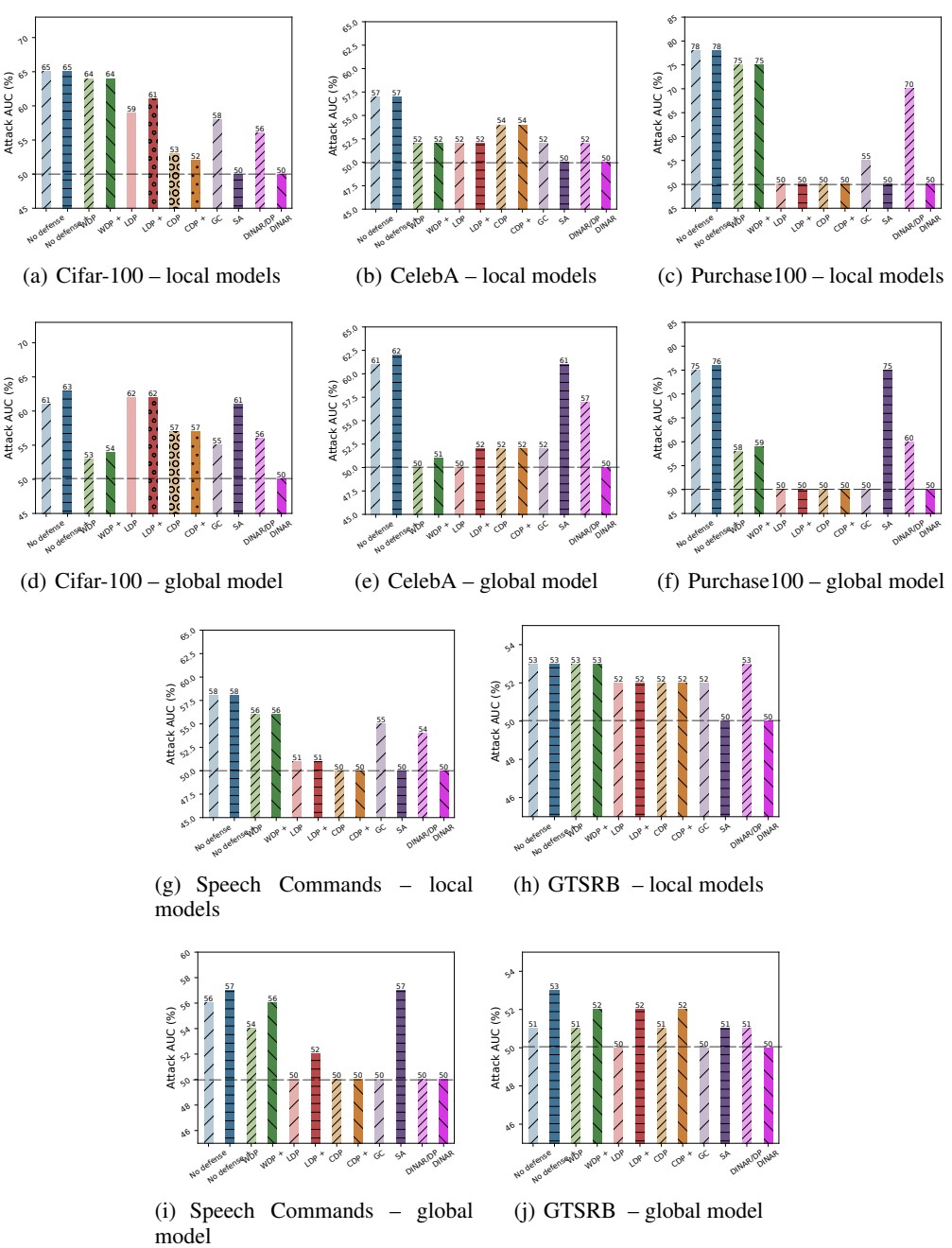

(a) Cifar-100 – local models

(b) CelebA – local models

(c) Purchase100 – local models

(d) Cifar-100 – global model

(e) CelebA – global model

(f) Purchase100 – global model

(g) Speech Commands – local models

(h) GTSRB – local models

(i) Speech Commands – global model

(j) GTSRB – global model

Figure 8: Privacy leakage with DINAR and state-of-the-art protection mechanisms with Cifar-10–
The horizontal dashed line represents the optimal value of attack AUC (50%)

## E APPENDIX – ADDITIONAL EVALUATION OF PRIVACY PROTECTION

In the following, we provide extended results for the experimental scenario described for Cifar-10
in §5.2, for all considered datasets. We provide our results in Figure 8 for other datasets, showing
a similar tendency; Differential Privacy based solutions provide a visible protection but limited in
some cases, in particular on local models. For instance, even if LDP and CDP reduce the Attack
AUC to 50% for the Purchase100 dataset, they still struggle in the cases of Cifar-100, CelebA,
GTSRB and Speech Commands. In exchange, DINAR provides the best privacy protection in all

cases, by reducing the attack AUC to 50%, corresponding to the optimal protection value. These results confirm our analysis of the behaviour of DINAR on Cifar-10, that remains constant for all datasets. These results corroborate our initial observation that DINAR remains the most efficient protection technique in case the objective is to protect both local and global models.

## F APPENDIX – PRIVACY PROTECTION UNDER NON-IID FL SETTINGS

In the following, we consider different non-IID FL settings, and evaluate their impact on the actual privacy protection achieved by different protection methods. We vary the non-IID FL dataset distribution using the Dirichlet function (Kotz et al., 2004) and its $\alpha$ parameter. The lower the Dirichlet's $\alpha$ value is, the more non-IID FL distribution is. Figure 9 presents the results of the evaluation of different non-IID distributions of the GTSRB dataset, and compares the utility and the resilience of clients' models to membership inference attacks when different privacy protection methods are applied, as well as when no defense is applied. Overall, for all cases but DINAR, the lower the non-IID distribution is, the higher the attack success rate is since the membership inference shadow model is able to better learn on such data. In the case of DINAR, the privacy protection is independent from the underlying non-IID setting and remains minimal at 50%. When it comes to model utility, obviously, the lower the non-IID distribution is, the higher the model utility is, although, DINAR reaches the highest model accuracy when protecting the model.

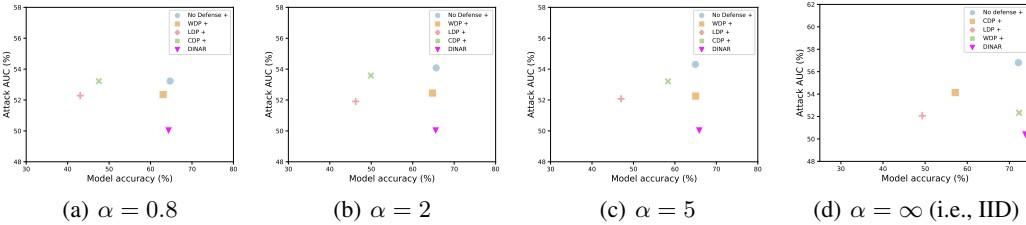

(a) $\alpha = 0.8$     (b) $\alpha = 2$     (c) $\alpha = 5$     (d) $\alpha = \infty$ (i.e., IID)

Figure 9: Privacy leakage vs. model utility under different non-IID FL settings – GTSRB dataset

## G APPENDIX – PRIVACY PROTECTION UNDER DIFFERENT NUMBERS OF CLIENTS

We evaluate the impact of different numbers of FL clients on the actual performance of DINAR. Figure 10 reports the attack AUC vs. the client model accuracy, comparing DINAR against the no defense baseline. In each case, the whole Purchase100 dataset was splitted into subsets for the different FL clients. Obviously, the fewer the clients are, the higher the client model accuracy is, since fewer clients implies more data per client. However, and independently from the number of clients, DINAR is able to counter MIAs with an attack AUC of 50%.

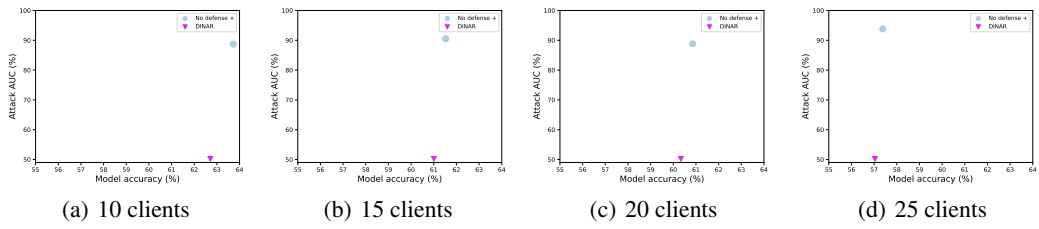

(a) 10 clients     (b) 15 clients     (c) 20 clients     (d) 25 clients

Figure 10: Privacy leakage vs. model utility under different numbers of FL clients – Purchase100

## H APPENDIX – PRIVACY PROTECTION OF MORE OR LESS VULNERABLE DATA SAMPLES

In the following, we evaluate the actual privacy achieved by different protection mechanisms by considering more specifically, on the one hand, data points that are easier to infer by the membership

inference attacker[5], and on the other hand, data points that are more difficult to infer by the attacker. Figure 11 presents the attack AUC vs. the client model accuracy of such an evaluation with the Purchase100 dataset. Obviously, the attack is more efficient on the most vulnerable data points with no defense, reaching an attack AUC approaching 100%. Privacy protection techniques based on gradient compression and WDP+ also provide less protection for more vulnerable data points, compared to less vulnerable ones. In contrast, DINAR as well as the other protection methods (i.e., secure aggregation, and other DP-based methods) provide good protection independently from the level of vulnerability of data points, although the former provides better model utility.

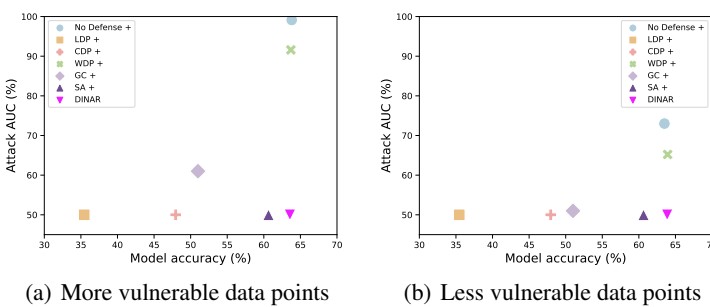

(a) More vulnerable data points      (b) Less vulnerable data points

Figure 11: Privacy leakage vs. model utility for more or less vulnerable data points – Purchase100

## I    APPENDIX – PRIVACY PROTECTION WITH DIFFERENT PRIVACY BUDGETS

In the following, we consider the recent PrivateFL-LDP differential privacy-based method (Yang et al., 2023), with different privacy budgets. We evaluate its actual resilience to membership inference attacks, and the model accuracy. We also compare PrivateFL-LDP against DINAR and against the case where no defense is applied. Figure 12 presents these results. Obviously, small privacy budgets provide better privacy. However, in order to reach the best privacy protection of 50%, PrivateFL-LDP drastically degrades the model accuracy to 13%. Whereas DINAR is able to keep a high model accuracy close to the no defense baseline, while effectively protecting against the MIAs.

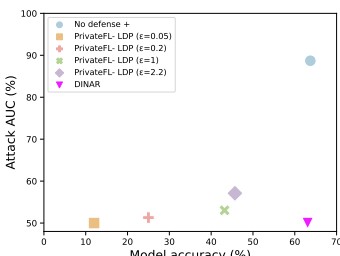

Figure 12: Privacy leakage vs. model utility under different DP privacy budgets – Purchase100

## J    APPENDIX – ABLATION STUDY

In order to evaluate the actual impact of the adaptive gradient descent in DINAR on the performance of the model, we report an ablation study where DINAR uses other state-of-the-art gradient descent optimizers, such as Adam (Kingma & Ba, 2015a), ADGD (Malitsky & Mishchenko, 2020), and AdaMax (Kingma & Ba, 2015a). Figure 13 presents the highest membership inference attack AUC against the models. Moreover, although not shown in the figure, all considered gradient descent techniques provide the same privacy protection level (i.e., an attack AUC of 50%).

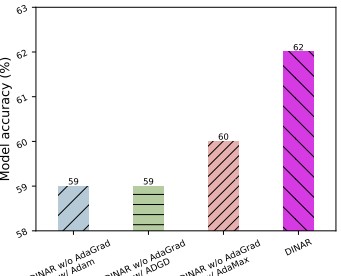

Figure 13: Ablation study – Purchase100

---

[5]Data points with a high inference confidence score by the attacker, i.e., a confidence score above the quantile of 0.8.

# K Appendix – Memory-Related Cost of FL Privacy-Preserving Mechanisms

Our study delves deep into the realm of GPU memory usage in privacy-preserving federated learning, unraveling captivating insights. Through meticulous analysis, we unveil the impact of various privacy mechanisms, including LDP, CDP, WDP, GC and SA on memory consumption during local model training. They show that in that case, running differential privacy algorithms increases the GPU Memory usage by 168% compared to a no defense scenario.

In exchange, DINAR does not introduce any computational comparable operation by definition, resulting in having no significant impact on GPU memory usage. Our findings paint a compelling picture, showcasing a systematic increase in GPU memory usage with the implementation of these privacy measures. First, the addition of calibrated noise, a fundamental technique in differential privacy, requires storing the noise values, which increases memory usage. Second, tracking and managing the privacy budget, which represents the maximum allowable privacy loss, necessitates additional memory to maintain the budget information. Lastly, the need for maintaining an aggregation buffer to collect model updates before applying privacy mechanisms adds to the memory requirements. This reasonably explains why DINAR is optimal from the perspective of GPU memory in comparison with differential privacy, as it doesn't involve noise addition nor privacy budget management.

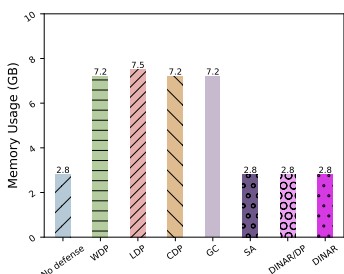

Figure 14: Cost of FL privacy-preserving mechanisms in terms of memory usage – GTSRB with VGG11

