# OpenReview forum: "DINAR: Fine-Grained Privacy Preserving Federated Learning"
_ICLR.cc/2024/Conference — Submitted to ICLR 2024_

### Official Review · Reviewer_nS4z · 2023-10-22

**Soundness:** 2 fair
**Presentation:** 2 fair
**Contribution:** 2 fair
**Rating:** 5
**Confidence:** 3

**Summary:**

The authors propose an approach for protecting FL neural network layers that leak more private information. This approach is motivated by an observation (Mo et al.; 2021) that is there is a layer in neural networks that leaks more private information than other layers.

In order to compensating accuracy, adaptive gradient descent is used. The evaluation results show that the proposed idea reduces the membership inference attack success rate with good model accuracy.

**Strengths:**

+ The idea seems to be interesting.
+ The authors conducted extensive experiments for comprehensive analysis.

**Weaknesses:**

- I have privacy concerns for other layers.
- The motivation seems interesting, but it is from an unpublished paper.
- There is no discussion of similar related works.

**Questions:**

1. The authors aim to protect a specific layer in FL models. What are the privacy risks contained in other layers? I think this protection is insecure.
2. The motivation is from an unpublished paper. Are there any similar papers from reputable conferences/journals? I have great concerns about the reliable analysis of the motivation. Although we should care about both published and unpublished papers, I think it would be better for authors to find more support for their conclusive motivation.
3. the authors criticize differential privacy (DP) and cryptography in the related works. The authors said they are different and novel. However, given my understanding, the proposed idea is less secure than DP and cryptography-based works. I do not think the comparison is fair.
4. The authors missed the discussion of similar related works in the same research direction.
5. The author claims better model accuracy. I suspect it is from using adaptive gradient descent. If so, the improved utility is not from the newly-designed protection, i.e., not novel. Where is the high utility from?

**Details Of Ethics Concerns:**

N.A.

---

> ### Author Response · Authors · 2023-11-15
>
> **What are the privacy risks contained in other layers?:**
>
> [Mo et al.; 2021] and Fig 1 compare the risk of information leakage of different layers of several datasets and model architectures. Protecting specifically the most sensitive layer also shows the best privacy protection for the overall model, as shown in Fig 3, where the fine-grained obfuscation was applied on the one or the other of the model layers.
>
> **Similar related works in the same research direction:**
>
> In addition to DP-based privacy-preserving methods, comparison and evaluations with other related works based on gradient compression and secure aggregation were included to the paper.
>
> **The motivation is from an unpublished paper. Are there any similar papers from reputable conferences/journals?:**
>
> In addition to the study of [Mo et al., 2021], that was published in ICLR 2021 - volume Workshop on Distributed and Private Machine Learning, the following paper could also be cited:
> Fan Mo, et al. PPFL: privacy-preserving federated learning with trusted execution environments. ACM MobiSys 2021.
>
> **Where is the high utility from?:**
>
> Better model accuracy comes from the combination of the fine-grained approach that obfuscates specifically the sensitive layer and the adaptive gradient descent.
> For a fair comparison with related works in terms of model accuracy, all considered privacy protection methods are also evaluated with a variant that applies the adaptive gradient descent as well (cf. Figures 5 and 6, for instance LDP vs. LDP+ with the adaptive gradient descent).

---

> > ### Comment · Reviewer_nS4z · 2023-11-16
> >
> > Thanks to the authors for their response.

---

> > > ### Author Response · Authors · 2023-11-22
> > >
> > > Dear Reviewer,
> > >
> > > Thank you again for your comments and for your constructive suggestions.
> > >
> > > We have addressed the suggestions in the updated version of the paper, and we have described how the paper was modified accordingly, including among others:
> > > - new experimental results of non-IID FL settings
> > > - new experimental results with different numbers of FL clients
> > > - new experimental results with different DP privacy budgets
> > > - new experimental results specifically targeting vulnerable data points
> > > - comparison to more recent related works on methods protecting FL systems againts membership inference attacks, inluding DP-based techniques, cryptographic methods, and methods based on gradient compression
> > >
> > > Furthermore, the title of the paper has been replaced by the following to better reflect its content: Heuristic-Guided Privacy against Membership Inference Attacks in Federated Learning.
> > >
> > > Please let us know if the updates to the paper handle your comments and suggestions, or if further elements should be added.
> > >
> > > Best regards,

---

> > > > ### Comment · Reviewer_nS4z · 2023-11-23
> > > >
> > > > Thanks for showing more experimental results. I have improved my rating given the experiments.
> > > >
> > > > However, in Appendix Table 3, the comparison looks wrong... Differential privacy (DP) and cryptography provide different types of privacy, respectively. Informally, DP provides individual privacy (e.g., example-level or user-level); however, cryptography provides indistinguishability to randomness. Thus, I do not think the works can be marked as providing "model privacy" in the same manner using different tools. If privacy is not clearly differentiated, motivation/comparison is not convincing either.

---

> ### Author Response · Authors · 2023-11-23
>
> Dear Reviewer,
>
> Thank you for very much your prompt feedback, and for your kind message.
> We agree with you, in Appendix Table 3, the type of privacy should be better defined. The table wasd updated accordingly to specifically target protection against membership inference attacks in federated learning.
>
> Please let us know if these updates handle your comments, or if further elements should be added.
>
> Best regards,

---

### Official Review · Reviewer_AcVa · 2023-10-29

**Soundness:** 2 fair
**Presentation:** 3 good
**Contribution:** 3 good
**Rating:** 5
**Confidence:** 4

**Summary:**

The paper proposes DINAR, a method designed to enhance the privacy of Federated Learning (FL) systems, specifically guarding against membership inference attacks. DINAR employs a straightforward yet efficient fine-grained approach, focusing on protecting the most vulnerable model layer in terms of privacy. This approach ensures effective and non-intrusive privacy protection in FL. Additionally, DINAR addresses potential accuracy losses in the protected model by leveraging adaptive gradient descent, thus optimizing model utility.

**Strengths:**

1. This paper proposes a fine-grained defense against MIAs. The authors investigate the impact of MIAs on different layers and provide empirical insights about perturbing some specific layers rather than the whole model.

2. The authors conduct extensive experiments on multiple datasets.

**Weaknesses:**

1. The authors do not provide a theoretical privacy guarantee against MIAs. Without the theoretical guarantee, people cannot analyze the effectiveness and generalization of the proposed privacy defense method.

2. It is unclear how the server or the clients derive the sensitive layer, e.g., layer p. Do all the clients share the same sensitive layer even under non-IID settings? If the clients have different sensitive layers, how does Dinar solve the divergence problem of local model updates since the clients perturb different layers?

3. It is unclear which MIA methods Dinar is evaluated against in the paper. The author should claim the attack methods more explicitly.

4. It might be a good idea to shrink the scope of the title to "privacy against MIA" rather than general privacy-preserving FL.

**Questions:**

Please see the weaknesses.

---

> ### Author Response · Authors · 2023-11-15
>
> **Which MIA methods Dinar is evaluated against in the paper?:**
>
> DINAR and the other defense methods in the paper are evaluated against the MIA method proposed by [Shokri et al., 2017], where shadow training techniques are used to build an attack model whose purpose is to distinguish the target model’s behavior on the training inputs from its behavior on the inputs that it did not encounter during training. This will be better explained in the paper.
>
>
> **Do all the clients share the same sensitive layer?:**
>
> All FL clients know the sensitive layer, that is the same for all clients. This assumption is similar to the fact that all FL clients know the model architecture that is used for their data, which is the same for all clients.
> The knowledge of the sensitive layer is obtained thanks to a prior study [Mo et al.; 2021], or an empirical evaluation (cf. Fig. 1). Furthermore, an additional evaluation of the impact of various FL non-IID settings on the actual privacy protection is included.
>
>
> **Heuristics vs. optimization-based approaches:**
>
> The proposed protection method is indeed based on a heuristic. Although heuristics-based approaches do not provide strict guarantees, we believe that they are useful and sometimes necessary as a first step for practical solutions in many research problems and areas, such as artificial intelligence [1,2], medicine [3], law [4], etc.
> We believe that it is our responsibility as scientists to share these findings, for a better understanding of the problem and the behavior of a system under certain conditions.
> To better reflect the followed approach, the title of the paper will be replaced by: Heuristic-Guided Privacy against Membership Inference Attacks in Federated Learning.
>
> [1] C.-A. Cheng, et al. Heuristic-Guided Reinforcement Learning. NeurIPS 2021.
>
> [2] O. Salzman, et al. Heuristic-Search Approaches for the Multi-Objective Shortest-Path Problem: Progress and Research Opportunities. IJCAI 2023.
>
> [3] J. N. Marewski, et al. Heuristic decision making in medicine. Dialogues Clin. Neurosci.,14(1), 2012.
>
> [4] G. Gigerenzer, G., et al. Heuristics and the Law. Cambridge, MA, MIT Press, 2007.
>
> **Shrink the scope of the title to privacy against MIA:**
>
> Thank you for the suggestion. The title of the paper will be replaced by: Heuristic-Guided Privacy against Membership Inference Attacks in Federated Learning.

---

> > ### Comment · Reviewer_AcVa · 2023-11-21
> > **Reply to the rebuttal**
> >
> > Thanks to the authors for their rebuttal. I'd like to increase my score.

---

> > > ### Author Response · Authors · 2023-11-22
> > >
> > > Dear Reviewer,
> > >
> > > Thank you again for your comments and for your constructive suggestions.
> > >
> > > We have addressed the suggestions in the updated version of the paper, and we have described how the paper was modified accordingly, including among others:
> > > - new experimental results of non-IID FL settings
> > > - new experimental results with different numbers of FL clients
> > > - new experimental results with different DP privacy budgets
> > > - new experimental results specifically targeting vulnerable data points
> > > - comparison to more recent related works on methods protecting FL systems againts membership inference attacks, inluding DP-based techniques, cryptographic methods, and methods based on gradient compression
> > >
> > > Furthermore, the title of the paper has been replaced by the following to better reflect its content: Heuristic-Guided Privacy against Membership Inference Attacks in Federated Learning.
> > >
> > > Please let us know if the updates to the paper handle your comments and suggestions, or if further elements should be added.
> > >
> > > Best regards,

---

### Official Review · Reviewer_uPn1 · 2023-11-01

**Soundness:** 3 good
**Presentation:** 3 good
**Contribution:** 2 fair
**Rating:** 5
**Confidence:** 4

**Summary:**

The authors propose a privacy-preserving method for FL, called  DINAR. Specifically, the paper presents an empirical analysis of how much each layer of a neural network leaks membership privacy information and identifies the most privacy-sensitive layer. Then, it proposes a fine-grained approach that obfuscates the most privacy-sensitive layer of the model, before sending it to the server for aggregation, and restores it at the client-side for personalization. It also adopts adaptive gradient descent for local training to improve the utility of the protected model. Finally, it evaluates DINAR with six datasets and four neural networks, and compares it with three FL privacy protection mechanisms. It shows that DINAR achieves effective privacy protection, without hurting model accuracy or inducing computational overhead.

**Strengths:**

- Originality: The proposed DINAR is new in its approach to obfuscating the most privacy-sensitive layer of the model before sending it to the server for aggregation, and restoring it at the client-side for personalization.
- Quality: The quality of the paper is good in its thorough empirical analysis of layer-wise privacy characterization and obfuscation.
- Clarity: The paper is well-written and clear in its presentation. The methodology is explained in detail, making it easy for readers to understand.
- Significance: By identifying and protecting the most privacy-sensitive layer of a model, DINAR can potentially help to advance FL techniques that balance privacy protection with model accuracy.

**Weaknesses:**

Novelty
-  It would be helpful if the authors could provide a clear and explicit comparison between the proposed DINAR method and the layer-wise privacy characterization studied in FL by Mo et al. (2021). Discussing the differences, similarities, and potential advantages of DINAR over Mo et al. will enhance the reader's understanding of the novel contributions of the present study.

Related Work
- The literature review on DP & FL seems outdated, with references primarily from 2019 and 2020. Recent advancements have significantly improved the utility of DPFL algorithms. Therefore, it is recommended that the authors conduct a comprehensive review of the latest DP & FL algorithms,  such as [1,2,3,4].

Clarifications:
- Generalization Gap of the layers: “The generalization gap of the penultimate layer is notably higher than the generalization gap of the other layer” This conclusion about the “penultimate layer” could be specific to certain model architectures used in Figure 1.  Can the authors clarify the extent to which this claim holds true across diverse model architectures?

- Adaptive Gradient Descent vs. Adam: Could the authors shed more light on why adaptive gradient descent exhibits superior convergence behavior compared to Adam?   “Given the high-dimensional nature of optimization problems in neural networks, this technique dynamically adjusts the learning rate for each dimension in an iterative manner.”  this statement holds not only for adaptive gradient descent but also for Adam.


Baselines:
- Comparison to DP-based Techniques:  My concern is that the comparison to DP-based techniques in Section 4.4 Figure 4 might be unfair, as the specific $\epsilon$ used for DP methods is either undisclosed or excessively large. According to Section 5.1,  $\epsilon$ might be set to 2.2 for all DP-related experiments throughout the paper.  Note that DP, by definition, safeguards data privacy against membership inference attacks. The suboptimal empirical privacy performance of DP methods in Figure 4 could potentially be attributed to the utilization of a large privacy budget. It would be more convincing if the authors could evaluate MIA against DP methods under a small privacy budget.
- Comparing the proposed method with recent DP & FL techniques, which have state-of-the-art  DP utility,  would strengthen the submission, which helps to highlight the advantages and unique contributions of the proposed method.

Experiments:
  - It is claimed that “leveraging adaptive gradient descent and, thus, further maximizing model utility”. However, there are no ablation studies or analyses verifying the effectiveness of adaptive gradient descent, compared to other optimization methods such as Adam and SGD.

  - The experimental setup appears to be overly simplistic, with only 5 clients considered for all datasets. This is contrary to the typical cross-silo and cross-device settings, which usually involve a significantly larger number of clients.
  - Additionally, partitioning the entire dataset among just 5 clients might result in each client possessing a sufficient quantity of data, thereby ensuring the local model is well-trained and potentially leads to memorization or overfitting phenomena for specific layers. It might be helpful to explore how the effectiveness of the proposed method under MIA attacks might be influenced by varying the number of clients, the size of local data, or the number of local training epochs.
  - Some experiment details are not provided.  What is the number of local epochs for each FL client? How many FL rounds are trained for all methods?
  - Model Architecture: “we consider the CelebA dataset with a model containing eight convolutional layers.” Does this mean that the authors trained a classification model exclusively using convolutional layers? There should be at least one fully connected layer to predict the class.

Lack of theoretical guarantee:
- While the proposed method is insightful, it is primarily based on heuristics and lacks privacy guarantees. Consequently, It is possible that advanced membership inference attacks could compromise the proposed method, whereas DP, with a small $\epsilon$, is guaranteed to provide privacy preservation.

Typos:
-  There is a missing period in the  caption of Figure 1.

Reference:
- [1] Projected federated averaging with heterogeneous differential privacy. VLDB 2021.
- [2] On privacy and personalization in cross-silo federated learning. NeurIPS 2022.
- [3] Make Landscape Flatter in Differentially Private Federated Learning. CVPR 2023.
- [4]  PRIVATEFL: Accurate, Differentially Private Federated Learning via Personalized Data Transformation. USENIX 2023.

**Questions:**

Please see the questions in "Weaknesses".

---

> ### Author Response · Authors · 2023-11-15
>
> **Comparison between DINAR and the layer-wise privacy characterization studied by Mo et al. (2021):**
>
> Whereas Mo et al. (2021) propose metrics to quantify information leakage from gradients computed over the training data, DINAR proposes a protection method for a fine-grained obfuscation of gradients of the sensitive layer that leaks more information. The sensitive layer was precisely identified thanks to a metric defined in Mo et al. (2021).
> This will be better clarified in the paper.
>
> **Sensitive layer of a model architecture:**
>
> Based on a prior knowledge of the layer sensitivity of a model architecture, e.g., through a prior study [Mo et al.; 2021], or an empirical evaluation (cf. Fig. 1), DINAR algorithm’s parameter p is configured with the index of the model sensitive layer.
>
> **More recent DP/FL  literature:**
>
> Thank you for the additional references. As suggested, more recent related works on DP in FL are now considered and compared against the other methods. We also added recent related works that make use of other privacy protection techniques such as gradient compression. The paper has been updated accordingly.
>
> **The specific used for DP methods is either undisclosed or excessively large + Evaluate MIA against DP methods under a small privacy budget:**
>
> The privacy budget of DP methods is indeed described in §5.1, when introducing the baselines.
> In addition, a new empirical evaluation considering different DP privacy budgets was included to the paper, in order to evaluate the actual tradeoff between privacy and utility under different (more or less small) DP settings.
>
> **Adaptive Gradient Descent vs. Adam:**
>
> Section 4.3 has been updated to better clarify the similarities and differences between Adam and the considered adaptive gradient descent.
>
> **Ablation study:**
>
> Thank you for pointing out this. An ablation study is added to the paper, to show the improvement of model utility.
>
>
> **Only 5 FL clients:**
>
> A new empirical evaluation with various numbers of FL clients is added to the paper.
>
> **Additional experiment details should be provided:**
>
> In our experiments, we consider 5 local epochs for each FL client, and 50 (resp. 80 and 300) FL rounds for Cifar-10, Cifar-100, GTSRB, CelebA (resp. SpeechCommands, and Purchase 100). This will be clarified in the paper.
>
> **CelebA model architecture:**
>
> For the CelebA dataset, we consider a model that includes eight convolutional layers and one fully connected layer. This will be clarified in the paper.
>
> **Heuristics vs. optimization-based approaches:**
>
> The proposed protection method is indeed based on a heuristic. Although heuristics-based approaches do not provide strict guarantees, we believe that they are useful and sometimes necessary as a first step for practical solutions in many research problems and areas, such as artificial intelligence [1,2], medicine [3], law [4], etc.
> We believe that it is our responsibility as scientists to share these findings, for a better understanding of the problem and the behavior of a system under certain conditions.
> To better reflect the followed approach, the title of the paper will be replaced by: Heuristic-Guided Privacy against Membership Inference Attacks in Federated Learning.
>
> [1] C.-A. Cheng, et al. Heuristic-Guided Reinforcement Learning. NeurIPS 2021.
>
> [2] O. Salzman, et al. Heuristic-Search Approaches for the Multi-Objective Shortest-Path Problem: Progress and Research Opportunities. IJCAI 2023.
>
> [3] J. N. Marewski, et al. Heuristic decision making in medicine. Dialogues Clin. Neurosci.,14(1), 2012.
>
> [4] G. Gigerenzer, G., et al. Heuristics and the Law. Cambridge, MA, MIT Press, 2007.

---

> > ### Comment · Reviewer_uPn1 · 2023-11-20
> >
> > Thank you for your feedback. I'd like to keep my rating.

---

> > > ### Author Response · Authors · 2023-11-22
> > >
> > > Dear Reviewer,
> > >
> > > Thank you again for your comments and for your constructive suggestions.
> > >
> > > We have addressed the suggestions in the updated version of the paper, and we have described how the paper was modified accordingly, including among others:
> > > - new experimental results of non-IID FL settings
> > > - new experimental results with different numbers of FL clients
> > > - new experimental results with different DP privacy budgets
> > > - new experimental results specifically targeting vulnerable data points
> > > - comparison to more recent related works on methods protecting FL systems againts membership inference attacks, inluding DP-based techniques, cryptographic methods, and methods based on gradient compression
> > >
> > > Furthermore, the title of the paper has been replaced by the following to better reflect its content: Heuristic-Guided Privacy against Membership Inference Attacks in Federated Learning.
> > >
> > > Please let us know if the updates to the paper handle your comments and suggestions, or if further elements should be added.
> > >
> > > Best regards,

---

### Official Review · Reviewer_Y9fY · 2023-11-01

**Soundness:** 3 good
**Presentation:** 3 good
**Contribution:** 3 good
**Rating:** 5
**Confidence:** 4

**Summary:**

The paper proposes DINAR, a privacy-preserving FL method to defend against Membership inference attacks in FL by specifically hiding FL neural network layers that leak more private information than other layers from the FL server in the aggregation step of FL. In order to compensate for any potential loss in the accuracy of the protected model, DINAR combines the proposed approach with adaptive gradient descent.

**Strengths:**

- The proposed approach is simple and easy to apply in real-world scenarios of FL settings.
- The proposed method can reduce the attack success rate of membership inference attacks while maintaining high model performance.
- Extensive empirical experiments are conducted for analytical insights and evaluations.

**Weaknesses:**

- The privacy risk of Membership Inference Attacks is on the data-point level. Therefore, different data points will correlate to different neurons in different layers. As a result, the proposed method obfuscates a few layers and can not provide privacy protection for all data points in the clients' datasets.
- No guarantee is given for privacy protections.
- The considered attacks do not have high ASR in no defense models (e.g., 58% AUC in CelebA). Therefore, the defensive ability of the proposed method is degraded since it might be able to defend against weak attacks but not for stronger ones.

**Questions:**

1/ What is the attack success rate of the MIA on the vulnerable data points, i.e., the data points whose memberships are easy to infer?
2/ How does the proposed method deal with non-i.i.d problems in FL? Specifically, when the data is non-i.i.d, obfuscating multiple layers will incur a distribution shift between the client's local data and the data from other clients.
3/ In Figure 6, why the model with no protection can have lower model performance compared to the model trained by the proposed method?
4. The size of the images is very small and hard to read. I suggest the authors make it more straightforward for the audiences.

---

> ### Author Response · Authors · 2023-11-15
>
> **What is the attack success rate of the MIA on the data points whose memberships are easy to infer?**
>
> Thank you for pointing out this. An additional evaluation specifically targeting more vulnerable data points vs. less vulnerable ones, under different protection methods, has been added to the paper.  Overall, privacy protection techniques based on gradient compression and WDP+ provide less protection for more vulnerable data points, compared to less vulnerable ones. In contrast, DINAR as well as the other protection methods (e.g., secure aggregation, and other DP-based methods) provide good protection independently from the level of vulnerability of data points, although the former provides better model utility.
>
>
> **Non-IID settings:**
>
> A new experimental evaluation comparing privacy methods under different non-IID FL settings was conducted, and its results have been added to the paper. Overall, for all cases but DINAR, the lower the non-IID distribution is, the higher the attack success rate is, since the shadow model of the membership inference attacker is able to better learn on such data. In contrast, DINAR's privacy protection is independent from the underlying non-IID FL setting and remains minimal at 50%. When it comes to model utility, obviously, the lower the non-IID distribution is, the higher the model utility is, although, DINAR reaches the highest model accuracy when protecting the model.
>
> **In Fig 6, model with no protection can have lower model performance than model trained with DINAR, why?:**
>
> In Fig 6, No defense applies Adam, whereas No defense+ and DINAR both apply AdaGrad adaptive gradient descent which produce better model accuracy.
>
> **Heuristics vs. optimization-based approaches:**
>
> The proposed protection method is indeed based on a heuristic. Although heuristics-based approaches do not provide strict guarantees, we believe that they are useful and sometimes necessary as a first step for practical solutions in many research problems and areas, such as artificial intelligence [1,2], medicine [3], law [4], etc.
> We believe that it is our responsibility as scientists to share these findings, for a better understanding of the problem and the behavior of a system under certain conditions.
> To better reflect the followed approach, the title of the paper will be replaced by: Heuristic-Guided Privacy against Membership Inference Attacks in Federated Learning.
>
> [1] C.-A. Cheng, et al. Heuristic-Guided Reinforcement Learning. NeurIPS 2021.
>
> [2] O. Salzman, et al. Heuristic-Search Approaches for the Multi-Objective Shortest-Path Problem: Progress and Research Opportunities. IJCAI 2023.
>
> [3] J. N. Marewski, et al. Heuristic decision making in medicine. Dialogues Clin. Neurosci.,14(1), 2012.
>
> [4] G. Gigerenzer, G., et al. Heuristics and the Law. Cambridge, MA, MIT Press, 2007.
>
> **Attacks do not have high ASR in no defense models (e.g., 58% AUC in CelebA):**
>
> Although the MIA attack AUC in No defense on CelebA is 58%, it is much higher with other datasets such as Purchase 100, Cifar-10 and Cifar-100 where the attack AUC reaches respectively 80%, 70% and 65%.

---

> > ### Author Response · Authors · 2023-11-22
> >
> > Dear Reviewer,
> >
> > Thank you again for your comments and for your constructive suggestions.
> >
> > We have addressed the suggestions in the updated version of the paper, and we have described how the paper was modified accordingly, including among others:
> > - new experimental results of non-IID FL settings
> > - new experimental results with different numbers of FL clients
> > - new experimental results with different DP privacy budgets
> > - new experimental results specifically targeting vulnerable data points
> > - comparison to more recent related works on methods protecting FL systems againts membership inference attacks, inluding DP-based techniques, cryptographic methods, and methods based on gradient compression
> >
> > Furthermore, the title of the paper has been replaced by the following to better reflect its content: Heuristic-Guided Privacy against Membership Inference Attacks in Federated Learning.
> >
> > Please let us know if the updates to the paper handle your comments and suggestions, or if further elements should be added.
> >
> > Best regards,

---

### Author Response · Authors · 2023-11-15

Dear Reviewers,

Thank you for your comments and for your constructive suggestions.

We provide below our response to your questions.

We have addressed  the suggestions (are addressing them during the rebuttal period), and we describe in the following how the paper will be modified accordingly.
Please note that the title of the paper has been replaced by: Heuristic-Guided Privacy against Membership Inference Attacks in Federated Learning.

Best regards,

---

> ### Author Response · Authors · 2023-11-22
>
> Dear Reviewers,
>
> Thank you again for your comments and for your constructive suggestions.
>
> We provide below our response to your questions, and enclosed an updated version of the paper.
>
> We have addressed the suggestions, and we describe in the following how the paper was modified accordingly, including among others:
> - new experimental results of non-IID FL settings
> - new experimental results with different numbers of FL clients
> - new experimental results with different DP privacy budgets
> - new experimental results specifically targeting vulnerable data points
> - comparison to more recent related works on methods protecting FL systems againts membership inference attacks, inluding DP-based techniques, cryptographic methods, and methods based on gradient compression
>
> Furthermore, the title of the paper has been replaced by the following to better reflect its content: Heuristic-Guided Privacy against Membership Inference Attacks in Federated Learning.
>
> Please let us know if the updates to the paper handle your comments and suggestions.
>
> Best regards,

---

### Meta-Review · Area_Chair_VkLt · 2023-12-08

**Metareview:**

This paper proposes a new defense, DINAR, against membership inference attacks in the FL setting. DINAR uses heuristics such as model obfuscation and model personalization to reduce the effectiveness of MIAs while preserving model utility through adaptive training. Experiments on datasets such as CIFAR, CelebA and Purchase100, etc. show that DINAR can successfully mitigate existing MIAs with minimal loss of accuracy.

The main weakness that reviewers raised is regarding the paper's significance. In contrast to principled techniques such as LDP that provably prevent privacy leakage of any form, DINAR focuses on preventing (existing) MIAs using heuristics. Furthermore, the authors clarified in the response to Reviewer AcVa that only the shadow model attack [Shokri et al., 2017] was used for evaluation. This attack is very outdated and cannot capture the adversary's true capabilities. More recent attacks such as https://arxiv.org/abs/2112.03570 and https://arxiv.org/abs/2111.09679 should be used instead. AC recommends rejection based on these weaknesses.

**Justification For Why Not Higher Score:**

The approach lacks any form of principled privacy guarantee. As a purely empirical approach, the empirical evaluation is also weak, relying on out-of-date MIA attacks and a non-standard FL setup.

**Justification For Why Not Lower Score:**

N/A

---

### Decision · Program_Chairs · 2024-01-16

Reject